# Coping strategy among the women with metastatic breast cancer attending a palliative care unit of a tertiary care hospital of Bangladesh

Nashid Islam[1], A. K. M. Motiur Rahman Bhuiyan[1], Afroja Alam[1], Mostofa Kamal Chowdhury[1], Jheelam Biswas[1,2]*, Palash Chandra Banik[2], Md. Maruf Ahmed Molla[3], Mostofa Monwar Kowshik[4], Mridul Sarker[1], Nezamuddin Ahmed[1]

1 Department of Palliative Medicine, Bangabandhu Sheikh Mujib Medical University (BSMMU), Dhaka, Bangladesh, 2 Department of Non-Communicable Diseases, Bangladesh University of Health Sciences, Dhaka, Bangladesh, 3 Department of Microbiology and Immunology, SUNY Upstate Medical University, Syracuse, NY, United States of America, 4 Department of Medicine, Dhaka Medical College, Dhaka, Bangladesh

* jheelam.biswas@gmail.com

**Data Availability Statement:** All data relevant to the study are accessible in Mendely data, doi:10. 17632/95h2d6pdnj.1.

## Abstract

### Background

Breast cancer is one of the leading cancers among the Bangladeshi women. Coping helps these patients to adjust with this life-changing disease. Each individual has unique and different coping mechanism. But we know a little regarding their coping strategies. This study aims to explore the different coping strategies adopted by the women with metastatic (stage IV) breast cancer attending the palliative care unit and their relationship with the common mental health issues like anxiety and depression.

### Methods

This cross-sectional study was conducted among 95 patients with metastatic (stage IV) breast cancer attending the Department of Palliative Medicine, Bangabandhu Sheikh Mujib Medical University, Bangladesh from April 2021 to September 2021. Data was collected by face-to-face interview using a structured questionnaire adapted from Hospital Depression and Anxiety Scale (HADS), Brief COPE inventory and Eastern Cooperative Oncology Group (ECOG) performance scale. Pearson correlation test was used to find the relationships between various domains of coping strategies and psychological variables. Correlation matrix was done to observe the internal correlation among different coping strategies. Kruskal-Wallis H test was done to find the relationship between different coping strategies and ECOG performance status.

### Result

The mean age of the respondents was 48.9 ± 9.9 years. Most of them were married (94.7%), Muslim (92.6%) and homemakers (82.1%). Commonly used coping strategies by

**Funding:** This study was funded by Bangabandu Skeikh Mujib Medical University, Dhaka, Bangladesh (Grant number: BSMMU/2021/3447) awarded to Nashid Islam. The funders have no involvement in the study design; collection, analysis, and interpretation of data; writing of the report; and the decision to submit the report for publication."

**Competing interests:** The authors have declared that no competing interests exist.

the patients were: acceptance (median 10; IQR 10), religion (median 9; IQR 8–10) and instrumental support (median 9; IQR 6–10). Significantly strong positive correlation was found between emotional and instrumental support (R = 0.7; p = 0.01), planning, acceptance and active coping (R = 0.7; p = 0.01); behavioral disengagement, self distraction and denial (R = 0.5; p = 0.01). Significantly fair negative correlation was observed between active coping and depression (R = -0.4, p <0.001). Patients with better performance status on ECOG scale (Grade 0–2) leaned more on the positive coping strategies like instrumental support, emotional support, positive reframing and venting.

## Conclusion

Different coping strategies, especially positive coping helps the patients to adapt with their disease over time. All women suffering from breast cancer should be routinely screened and assessed for psychological distress and ensure early intervention and management to promote a better quality of life.

## Background

Cancer is one of the leading causes of death among the non-communicable diseases globally. Breast cancer is the most common malignancy among women around the world with an estimated 2.3 million new cases diagnosed in the year 2020. Among them, 6% have metastatic breast cancer during the first diagnosis [1, 2]. Breast cancer has been reported to be the highest prevalent (about 19.3 per 100000 women) malignancy among the Bangladeshi women between 15 to 44 years of age. Many of these patients present at the late stage mostly because of social stigmata and lack of awareness. Unfortunately, there is no national cancer registry in Bangladesh, so the exact number of these patients remains unknown [3]. Diagnosis of metastatic breast cancer is a great shock for the patients. Its treatment and side effects have tremendous social and psychological impacts on them [4]. These patients experience intense stress over the period of illness, largely due to increasing physical symptom burden, emotional distress, body image disturbance, and disrupted daily activities. A study shows that along with the various physical symptoms these patients suffer from different psychological symptoms such as depression, anxiety and melancholy [5]. They also struggle with fear of death and spiritual issues [6].

When a woman is diagnosed with breast cancer she goes through certain stages of psychological responses. To adjust to this life-changing disease and its effects they often adopt some psychological maneuvers which are referred as coping [7].

Coping is a "process by which a person deals with stress, solves problems or makes decisions" [8]. Patients with advanced cancer and their families frequently go through a great deal of stress, and they adopt varieties of methods to cope. Each individual reacts to challenges differently and copes uniquely.

Different studies shows that, women with breast cancer who have a 'fighting spirit' have better chance of survival than women who are compliant or experience helplessness and hopelessness [9, 10]. Several coping strategies can be used in such stressful conditions, and the strategies implemented depend on both the individual's cognitive appraisal of the situation and his/her emotional status [11]. Some authors categorized these strategies as emotion-focused and problem-focused, whereas others classified them as active and avoidant strategies [12].

Coping strategies adopted by cancer patients are not fixed in stone. It changes with the trajectory of the disease. As the disease progresses, many of these patients lean on acceptance and problem-focused coping, but some of them adopt negative coping like denial. Such coping strategies are associated with multiple outcomes, including impact on quality of life, depression and anxiety symptoms, understanding of disease prognosis, and care at the end of life [13].

Palliative care is an approach that improves the quality of life of patients and their families facing the problems associated with life-threatening illness. Studies demonstrate that palliative care integrated with oncology care, not only improves outcomes but also enhances coping in patients with advanced cancer. Since coping strategies are modifiable, palliative care aims to help the patients with advanced cancer adopting more positive coping strategies and improving their symptom burden, mental health, treatment outcome and quality of life [13]. Early consultation with palliative care specialists also helps in lowering anxiety and depression among these patients [14].

Since the initiation of palliative care in Bangladesh, Department of Palliative Medicine of Bangabandhu Sheikh Mujib Medical University (BSMMU) has been providing this care to the cancer patients and their families. According to their database almost 28% of these patients are suffering from metastatic (stage IV) breast malignancy (manually calculated). The goal of providing palliative care to these patients is not only to treat their physical symptoms, but also offering psychological, social and spiritual support. But we know a little regarding the adoption of coping strategies among these patients, which hinders us from offering them necessary support to cope with the disease and decrease their sufferings. So, this study aims to explore the different coping strategies adopted by the women with metastatic (stage IV) breast cancer attending our department as well as their relationship with the common mental health issues like anxiety and depression, and to our best of knowledge, it is the first study to do so.

## Methods

### Study design and setting

This cross-sectional study was conducted among the female patients diagnosed with metastatic (stage IV) breast cancer attending the indoor and outdoor facilities of the Department of Palliative Medicine, Bangabandhu Sheikh Mujib Medical University (BSMMU), Shahbag, Dhaka. Data collection was done from April 2021 to September 2021.

### Sample criteria

Patients having stage IV breast cancer above 18 years of age, either admitted in the palliative care ward or attended the outdoor facility of the Department of Palliative Medicine, BSMMU were included in our study. Those patients, who have any disorders like stroke, motor neuron disease etc that can clearly interfere with cognition, were excluded.

### Sample size

We used census method to determine the sample size of the study. All the stage IV breast cancer patients either admitted in the palliative care ward or attended the outdoor facility of the Department of Palliative Medicine, BSMMU during the study period were listed from the hospital register. According to the hospital register, total 103 stage IV breast cancer patients visited our department (manually calculated) during the study period. Among them, 27 patients were admitted in the palliative care ward, and 76 received treatment form the outdoor. Six of them

had pre-existing cognitive impairment, and 2 of them refused to give consent. So our final sample size was 95.

### Data collection procedure

Data were collected by the principal investigator from both indoor and outdoor using a structured questionnaire in four parts. The first part contained the socio-demographic variables, disease status, and treatment-related variables.

The second part was used to assess the Performance Status of the patient by the "Eastern Cooperative Oncology Group (ECOG)" performance scale. The performance status is divided into five grades ranging from '0' to 'IV'. Grade 0 refers to the patients who are completely asymptomatic, fully active, and are able to carry on all pre-disease activities without restriction. Grade I refers to the patients who are symptomatic but completely ambulatory, having restricted in physically strenuous activity but are able to carry out work of a light or sedentary nature (e.g., light housework, office work). Grade II refers to the patients who are symptomatic, spend<50% time in bed during the day, are ambulatory and capable of all self-care but unable to carry out any work activities; also remain awake more than 50% of waking hours. Grade III refers to the patients who are symptomatic, spend >50% time in bed, but not bed-bound, are capable of only limited self-care and confined to bed or chair 50% or more of their waking hours. Grade IV refers to the patients who are completely bedbound and disabled. Grade V refers to death of the patient.

The third part contained the validated Bangla version of the 'Hospital Depression and Anxiety Scale (HADS)' questionnaire [15]. The part contained seven items that assess anxiety and seven items that assess depression. HADS-A or HADS-D score of 8 was defined as a case, score from 8 to 10 as mild, from 11 to 14 as moderate, and above 14 denotes as severe cases.

The fourth part of the questionnaire contained the validated Bangla version of the 'Brief COPE inventory' which was used to assess the coping strategies of the study subjects [16]. It is a multidimensional measure and presents fourteen scales all assessing different coping dimensions. This questionnaire includes 28 items that explore the following 14 coping strategies: self-distraction (items 4 and 22), active coping (items 2 and 10), denial (items 5 and 13), substance use (items 15 and 24), use of emotional support (items 9 and 17), use of instrumental support (items 1 and 28), behavioral disengagement (items 11 and 25), venting (items 12 and 23), positive reframing (items 14 and 18), planning(items 6 and 26), humor (items 6 and 26), acceptance (items 3 and 2), religion (items 14 and 18), and self-blame (items 8 and 27). After pre-testing the questionnaire among 10 cancer patients necessary corrections were made and the instrument was finalized.

Data was collected through face-to-face interviews. The most appropriate response was noted in the questionnaire by the researcher. When the patient was unable to follow a question or response verbally, the caregiver was requested to give proper information. Sensitive questions were asked privately, and also were allowed to write down (if a patient was uncomfortable discussing them openly). The duration of each interview was 30 minutes to 1 hour. Two to three patients were interviewed each day. Very frail patients were given multiple visits to complete an interview.

### Data analysis

All statistical analyses were performed using the SPSS version 26. Descriptive analysis was done for the categorical variables such as age, monthly family income, educational status, Eastern Cooperative Oncology Group (ECOG) performance status and treatment history. The

scores of each coping strategy of the Brief COPE scale were expressed in median and inter-quartile range (IQR).

The relationships between various domains of coping strategies and other psychological variables like depression and anxiety were studied by using the Pearson correlation test. Correlation matrix was done to see the correlation among different coping strategies. Relationship between different coping strategies and Eastern Cooperative Oncology Group (ECOG) performance status were assessed by Kruskal-Wallis H test. P value<0.05 was considered to be statistically significant.

### Ethical considerations

The ethical approval (Approval no: BSMMU/2021/3447, date: 17/04/2021) was obtained from the Institutional Review Board (IRB) of Bangabandhu Sheikh Mujib Medical University, Bangladesh. Written informed consent was taken from all the eligible patients. Sensitive questions were discussed privately. As they were terminally ill patients, their health conditions were considered during data collection.

## Results

The mean age of the patients enrolled in the study was 48.9± 9.9 years. The majority of them were married (94.7%), Muslim (92.6%), homemakers (82.1%) by profession, and had education above secondary level (63.2%). More than half of them (68.4%) had no known family history of breast malignancy. Most of them received disease-modifying treatments like surgery (87.4%), radiotherapy (70.5%), and chemotherapy (96.8%). Almost half (54.7%) of the patients received alternate therapies like homeopathy and herbal medications. More than half (52.6%) of the patients were symptomatic but completely ambulant (ECOG grade II) and a few (6.3%) patients were completely bed-bound (ECOG grade IV) (**Table 1**).

Nearly half of the patients (47.4%) were found to have no anxiety according to the Hospital Anxiety and Depression Scale (HADS), while four out of ten (44.2%) of the patients were considered as moderate to severely anxious.

Again, more than half (51.6%) patients had no depression while one out of ten (11.5%) patients were suffering from mild, three out of ten (36.9%) were suffering from moderate to severe depression (**Table 2**).

Among the different pattern of coping strategies adopted by the patients the most commonly used strategies were- acceptance (median 10; IQR 10), religion (median 9; IQR 8–10), and instrumental support (median 9; IQR 6–10) closely followed by emotional support (median 8; IQR 7–10), active coping (median 8; IQR 7–10), planning (median 8, IQR 8–10) and venting (median 8; IQR 6–10). The lowest used coping strategies were humor (median 2; IQR 2–3) and substance use (median 2; IQR 2) followed by denial (median 3; IQR 2–7) and behavioral disengagement (median 3; IQR 2–5) (**Table 3**).

Significantly strong positive correlations were found between emotional and instrumental support (R = 0.7; p = 0.01), planning and active coping (R = 0.7; p = 0.01), acceptance and active coping (R = 0.5; p = 0.01), denial and self distraction (R = 0.6; p = 0.01), behavioral disengagement and denial (R = 0.5; p = 0.01), venting, emotional and instrumental support (R = 0.7; p = 0.01). Significantly strong negative correlations were observed between active coping, planning and denial (R = -0.6; p = 0.01), behavioral disengagement, acceptance and planning (R = -0.5; p = 0.01) (**Table 4**).

Significantly fair negative correlation (r = -0.4, p<0.001) was found between active coping and depression which meant that those who adopted this coping strategy suffered less from

**Table 1. Socio-demographic characteristics of the patients (n = 95).**

| Socio-demographic characteristics | Frequency (n) | Percentage (%) |
|---|---|---|
| **Age (in years)** | | |
| Up to 40 | 21 | 22.1 |
| 41–50 | 37 | 38.9 |
| 51–60 | 24 | 25.3 |
| >60 | 13 | 13.7 |
| **Educational status** | | |
| No formal education | 15 | 15.8 |
| Up to primary | 12 | 12.6 |
| Secondary | 8 | 8.4 |
| Above secondary | 60 | .63.2 |
| **Occupational status** | | |
| Homemakers | 78 | 82.1 |
| Service holder | 17 | 17.9 |
| **Income group (in BDT)\*** | | |
| Lower middle (8,000–30,0000) | 28 | 29.5 |
| Upper middle (31,000–92,000) | 24 | 25.3 |
| High (>92,000) | 21 | 22.1 |
| **Marital status** | | |
| Married | 90 | 94.7 |
| Single | 5 | 5.3 |
| **Religion** | | |
| Islam | 88 | 92.6 |
| Hinduism and Christianity | 7 | 7.4 |
| **Family history of breast malignancy** | | |
| Yes | 30 | 31.6 |
| No | 65 | 68.4 |
| **Mode of treatment along with palliative care** | | |
| Chemotherapy | 92 | 96.8 |
| Radiotherapy | 67 | 70.5 |
| Surgery | 83 | 87.4 |
| Hormone therapy | 15 | 15.8 |
| Alternate therapy | 52 | 54.7 |
| **Eastern Cooperative Oncology Group (ECOG) performance status** | | |
| Grade 0 | 15 | 15.8 |
| Grade 1 | 50 | 52.6 |
| Grade 2 | 15 | 15.8 |
| Grade 3 | 9 | 9.5 |
| Grade 4 | 6 | 6.3 |

\*According to World Bank, 2021.

**Table 2. Anxiety and depression among the patients (n = 95).**

| Categories | Anxiety | Depression |
|---|---|---|
| | **No of respondents, n (%)** | |
| No | 45 (47.4) | 49 (51.6) |
| Mild | 8 (8.4) | 11 (11.5) |
| Moderate | 27 (28.4) | 15 (15.8) |
| Severe | 15 (15.8) | 20 (21.1) |

**Table 3. Coping strategies adopted by the patients (n = 95).**

| Coping strategies | Median score* | Interquartile range (IQR) |
|---|---|---|
| Instrumental support | 9 | 6–10 |
| Emotional support | 8 | 7–10 |
| Active coping | 8 | 7–10 |
| Planning | 8 | 8–10 |
| Acceptance | 10 | 10 |
| Self-distraction | 5 | 3–6 |
| Denial | 3 | 2–7 |
| Humor | 2 | 2–3 |
| Self-blaming | 4 | 2–5 |
| Behavioral disengagement | 3 | 2–5 |
| Venting | 8 | 6–10 |
| Positive reframing | 6 | 5–7 |
| Substance use | 2 | 2 |
| Religion | 9 | 8–10 |

*The higher the score, the more commonly used strategy.

depression. Meanwhile, there was no correlation found between coping strategy and anxiety (**Table 5**).

A significant relationship (p<0.05) was found among the median scores of instrumental support, emotional support, behavioral disengagement, venting and positive reframing with the patients' ECOG performance status. The patents with better performance status (ECOG 0-II) leaned more on the positive coping strategies like instrumental support, emotional support, venting and positive reframing. Again negative copings like behavioral disengagement denial and self-distraction were observed among the patients with poorer performance status (ECOG III- IV) (**Table 6**).

## Discussion

Breast cancer is one of the leading causes of cancer-related morbidity worldwide. Women with advanced breast cancer go through many psychological responses such as depression, anxiety, and sadness with which they need to cope on daily basis. This is the first study done in Bangladesh to explore the coping strategies among the Bangladeshi women with metastatic breast cancer getting palliative care.

In our study, acceptance is the most commonly used coping strategy by the patients with metastatic breast cancer. Acceptance teaches individuals to live with the reality of a difficult circumstance and accept the consequences of the illness's progression and adversity. As a result, we may conclude that patients are not accusing themselves for the onset/cause of their disease and are taking full responsibility for their current predicament. Carver et al. also found that acceptance is the most commonly used coping mechanism, alongside positive reframing and religion [17]. Another study from North India observed acceptance as the most common coping strategies for women with breast cancer with co-morbid depression [18].

Religion is another frequently adopted coping strategy among the women with advanced breast cancer. It is an emotion-focused strategy that involves mental activities that assist an individual in emotionally separating themselves from the stressor than bringing about any change in the environment. In these patients' minds life is planned by God and this illness is an undesirable event that they must accept. The breast cancer patients also claim that their

**Table 4. Correlation matrix of the coping strategies adopted by the patients (n = 95).**

R value (p value)

| Coping strategies | Instrumental support | Emotional support | Active coping | Planning | Acceptance | Self distraction | Denial | Humor | Self blaming | Behavioral disengagement | Venting | Positive reframing | Substance use | Religion |
|---|---|---|---|---|---|---|---|---|---|---|---|---|---|
| Instrumental support | 1 | | | | | | | | | | | | | |
| Emotional support | **0.7** **(0.01)** | 1 | | | | | | | | | | | | |
| Active coping | 0.3 (0.01) | 0.1 (0.99) | 1 | | | | | | | | | | | |
| Planning | 0.3 (0.01) | 0.1 (0.14) | **0.7** **(0.01)** | 1 | | | | | | | | | | |
| Acceptance | 0.3 (0.01) | 0.2 (0.01) | **0.5** **(0.01)** | 0.4 (0.01) | 1 | | | | | | | . | | |
| Self distraction | -0.2 (0.02) | -0.2 (0.01) | -0.3 (0.01) | -0.1 (0.21) | -0.1 (0.18) | 1 | | | | | | | | |
| Denial | -0.3 (0.01) | -0.3 (0.01) | **-0.6** **(0.01)** | **-0.5** **(0.01)** | -0.3 (0.01) | **0.6** **(0.01)** | 1 | | | | | | | . |
| Humor | -0.1 (0.31) | -0.1 (0.25) | -0.1 (0.49) | -.01 (0.97) | 0.1 (0.62) | 0.2 (0.01) | 0.2 (0.01) | 1 | | | | | | |
| Self blaming | -0.1 (0.21) | -0.1 (0.47) | 0.1 (0.63) | 0.3 (0.97) | 0.1 (0.71) | 0.3 (0.71) | -0.2 (0.84) | 0.4 (0.01) | 1 | | | | | |
| Behavioral disengagement | -0.2 (0.01) | -0.2 (0.05) | **-0.5** **(0.01)** | **-0.5** **(0.01)** | -0.2 (0.04) | 0.2 (0.01) | **0.5** **(0.01)** | 0.1 (0.06) | -0.1 (0.86) | 1 | | | | |
| Venting | **0.7** **(0.01)** | **0.6** **(0.01)** | 0.3 (0.01) | 0.3 (0.01) | 0.3 (0.01) | -.01 (0.08) | -0.2 (0.01) | -0.1 (0.57) | 0.1 (0.88) | -0.1 (0.05) | 1 | | | |
| Positive reframing | -0.1 (0.57) | -0.1 (0.78) | 0.2 (0.04) | 0.1 (0.01) | 0.1 (0.07) | 0.1 (0.11) | -0.1 (0.86) | 0.1 (0.29) | 0.1 (0.10) | 0.2 (0.78) | -0.1 (0.17) | 1 | | |
| Substance use | -0.1 (0.11) | -0.3 (0.01) | -0.1 (0.17) | -0.2 (0.81) | -0.03 (0.72) | 0.1 (0.14) | 0.2 (0.03) | 0.2* (0.01) | 0.03 (0.73) | 0.2 (0.03) | -0.07 (0.47) | -0.03 (0.72) | 1 | |
| Religion | 0.1 (0.11) | 0.05 (0.61) | 0.3 (0.01) | 0.2 (0.01) | 0.2 (0.01) | 0.1 (0.95) | 0.2 (0.80) | 0.1 (0.12) | -0.1 (0.43) | -0.1 (0.16) | 0.1 (0.24) | 0.1 (0.12) | -0.1 (0.07) | 1 |

**Table 5. Correlation of the coping strategies with depression and anxiety (n = 95).**

| Coping strategies | Depression | Anxiety |
|---|---|---|
| | R value (p value) | |
| Instrumental support | -0.2 (0.02) | -0.1 (0.07) |
| Emotional support | -0.2 (0.01) | -0.2 (0.01) |
| Active coping | **-0.4 (<0.001)** | -0.1 (0.06) |
| Planning | -0.2 (0.01) | -0.1 (0.34) |
| Acceptance | -0.2 (0.02) | -0.1 (0.09) |
| Self-distraction | 0.1 (0.26) | 0.1 (0.22) |
| Denial | 0.1 (0.06) | 0.1 (0.23) |
| Humor | 0.1 (0.33) | -0.1 (0.88) |
| Self-blaming | 0.1 (0.13) | 0.1 (0.24) |
| Behavioral disengagement | 0.2 (0.02) | 0.1 (0.22) |
| Venting | -0.2 (0.05) | -0.1 (0.45) |
| Positive reframing | -0.1 (0.15) | -0.1 (0.17) |
| Substance use | 0.1 (0.14) | 0.1 (0.20) |
| Religion | -0.2 (0.01) | -0.1 (0.65) |

Pearson correlation test was done.

belief in God calm them down and give them inner strength and bravery to fight the disease. They 'put everything in God's hands' in terms of death, the future, and uncertainty. Previous studies conducted in Egypt revealed that the women with breast cancer alleviate their fear regarding the uncertain future, disease recurrence, and death by 'leaving things to God's hands' [9]. Doumit et al. found that all Lebanese women, regardless of religious affiliation, thought cancer was something from God [19]. This thought helps the women accept their sickness because they cannot change something that came directly from God; they can only accept

**Table 6. Relationship between the coping strategies and ECOG score (n = 95).**

| Coping strategies | Grade-0 | Grade-I | Grade-II | Grade-III | Grade-IV | P value |
|---|---|---|---|---|---|---|
| | Mean rank | | | | | |
| Instrumental support | 50.13 | 56.51 | 33.13 | 28.33 | 38.42 | **0.004** |
| Emotional support | 45.07 | 57.63 | 35.93 | 30.33 | 31.75 | **0.003** |
| Active coping | 46.60 | *51.22* | *49.03* | 38.56 | 36.25 | 0.553 |
| Planning | 46.27 | *52.67* | *46.20* | 39.28 | 31.00 | 0.279 |
| Acceptance | 46.37 | 54.14 | 42.17 | 39.17 | 28.75 | 0.106 |
| Self-distraction | 36.97 | 45.66 | 54.53 | 67.39 | 49.67 | 0.080 |
| Denial | 41.60 | 44.30 | 55.33 | 59.31 | 59.83 | 0.194 |
| Humor | 47.17 | 50.04 | 49.40 | 44.72 | 34.50 | 0.550 |
| Self-blaming | 43.43 | 53.69 | 37.63 | 47.17 | 39.17 | 0.231 |
| Behavioral disengagement | 42.53 | 46.25 | 40.43 | 58.78 | 79.00 | **0.019** |
| Venting | 47.47 | 55.06 | 32.53 | 36.72 | 46.08 | **0.042** |
| Positive reframing | 40.87 | 50.03 | 60.67 | 45.50 | 21.00 | **0.022** |
| Substance use | 42.50 | 50.87 | 45.97 | 48.28 | 42.50 | 0.316 |
| Religion | 48.20 | 49.21 | 47.37 | 52.72 | 31.92 | 0.611 |

Kruskal-Wallis H test was done; Higher mean rank indicates more adoption of the respective strategy; p value <0.05 considered as significant.

it. The significant use of religion as a coping technique may explain why this group of patients' depression was mild to moderate in severity, as opposed to what one might predict as a reaction to cancer. Another qualitative phenomenological study done in Iran among the women with primary breast cancer diagnosis showed that patients believe that cancer is given by God (in a fatalistic viewpoint, which is unavoidable. It causes delay in starting treatment, and strategies to prevent recurrence or fighting cancer-related complications) [20].

Instrumental support is another commonly used strategy in to our study. Instrumental support means seeking advice from family members and friends. It has been observed that women who get social and emotional support tend to show less distress and go through less psychological turmoil [18]. Our group also uses active coping and planning that are problem-focused coping. These are comparatively less frequently used by the advanced cancer patients because these coping strategies are more useful in the early stage of diagnosis where the hope for cure can be fulfilled through careful planning and taking steps towards getting treatment. A study done among breast cancer patients on chemotherapy in Malaysia yielded the similar result [21]. We have also found a strong positive correlation among instrumental support, emotional support, planning, active coping and acceptance. It indicates that, those who have adopted those strategies have higher level of acceptance towards the disease. Also a significantly strong negative correlation has been observed between active coping, planning and denial. It means, those who are in less denial about their disease can actively cope with their situation. This evidence is supported by a study conducted in North India where the participants used these methods significantly more in the early stages of diagnosis. It has been theorized that, the women in the early stages of breast cancer are more prompt and employ planned problem solving to deal with the disease. Though the application of these coping mechanisms may generate stress for a short time, in the long run, they will begin to adjust to their situation [18].

Strategies like substance use, behavioral disengagement, and self-blame seem to have less impact among our participants. One possible reason may be that breast cancer has a favorable prognosis compared to many other types of malignancies. Women with breast cancer can go through treatment for a longer duration and try to deal with positive focused coping. But as the disease progresses, due to the severity of the symptoms caused by the metastasis, patients learn to adjust to it and accept the consequences. We have found a significantly strong negative correlation between acceptance and behavioral disengagement. It indicates that, those who have accepted their disease are less prone to behavioral disengagement. They believe that, this illness is their destiny and a God's test and surrender themselves. Therefore they accept their diagnosis and prognosis and do not blame themselves or adopt any dysfunctional way to cope.

A meta-analysis done in Sweden on coping with breast cancer concluded that, women with advanced breast cancer adopt disengagement coping more than those who have early-stage breast cancer which partially contradicts our study [22]. We have not explored the reasons behind this contraindication.

It is evident from our study that the women who adopt active coping suffer less from depression. On the other hand, there is no correlation found between coping strategy and anxiety. Some studies have shown that adaptive copers had lower levels of anxiety and depression than negative copers [23]. However, one study conducted in North India contradicts our findings. They have found a negative correlation between positive coping and depression. According to that study, the patients who are actively dealing their situation have psychological distress which manifests in the form of depression [18].

Positively focused coping strategies such as instrumental and emotional support, venting and positive reframing are more commonly adopted by the patients with better performance on the ECOG performance scale, while patients with poorer performance status seem to lean

more on the negative copings like behavioral disengagement, self-distraction and denial. Performance status is an essential prognostic factor for the survival of patients with breast cancer. A number of studies have stated that low-performance status leads to increased negative coping in the patients with cancer [24].

However, our study has several limitations. One limitation is that, this study only focuses on the coping strategies adopted at a single point of time, so it doesn't reflect the changes in those strategies along with the disease progression. In addition to that, some confounding factors have not been controlled. For instance, past psychiatric history or other psychological issues, which might precipitate anxiety and depression among these patients, are not explored. Another limitation is that, this study is conducted in a single center using a small, nonrandom purposive sampling technique, so the results cannot be generalized. For a better understanding of coping strategies adopted by advanced cancer patients, multi-center studies with greater sample size are required.

## Conclusion

In this study we have found that, the women with positive coping strategies suffer less from mental health problems. These strategies help people to adapt with their disease over time. Women with breast cancer should be encouraged to use positive coping strategies to ensure better adherence to treatment and also discourage negative strategies like denial, behavioral disengagement or self distraction which can delay their physical and psychological management.

## Supporting information

**S1 File.**
(DOCX)

## Author Contributions

**Conceptualization:** Nashid Islam, A. K. M. Motiur Rahman Bhuiyan, Afroja Alam, Mostofa Kamal Chowdhury, Jheelam Biswas, Mridul Sarker, Nezamuddin Ahmed.

**Data curation:** Nashid Islam, Afroja Alam, Mostofa Kamal Chowdhury, Jheelam Biswas, Palash Chandra Banik, Mostofa Monwar Kowshik, Mridul Sarker.

**Formal analysis:** Nashid Islam, A. K. M. Motiur Rahman Bhuiyan, Afroja Alam, Palash Chandra Banik, Mostofa Monwar Kowshik, Mridul Sarker.

**Funding acquisition:** Nashid Islam, A. K. M. Motiur Rahman Bhuiyan, Afroja Alam.

**Investigation:** Nashid Islam, A. K. M. Motiur Rahman Bhuiyan, Afroja Alam, Mostofa Kamal Chowdhury, Mridul Sarker.

**Methodology:** Nashid Islam, A. K. M. Motiur Rahman Bhuiyan, Afroja Alam, Mostofa Kamal Chowdhury, Jheelam Biswas, Palash Chandra Banik, Md. Maruf Ahmed Molla, Mostofa Monwar Kowshik.

**Project administration:** Nashid Islam, A. K. M. Motiur Rahman Bhuiyan, Afroja Alam, Mostofa Monwar Kowshik, Mridul Sarker.

**Resources:** Nashid Islam, A. K. M. Motiur Rahman Bhuiyan, Afroja Alam, Mostofa Kamal Chowdhury, Mostofa Monwar Kowshik, Mridul Sarker.

**Software:** Nashid Islam, Jheelam Biswas, Palash Chandra Banik, Mostofa Monwar Kowshik, Mridul Sarker.

**Supervision:** A. K. M. Motiur Rahman Bhuiyan, Afroja Alam, Mostofa Kamal Chowdhury, Palash Chandra Banik, Md. Maruf Ahmed Molla, Mridul Sarker, Nezamuddin Ahmed.

**Validation:** Nashid Islam, A. K. M. Motiur Rahman Bhuiyan.

**Visualization:** Nashid Islam, A. K. M. Motiur Rahman Bhuiyan, Mostofa Kamal Chowdhury, Nezamuddin Ahmed.

**Writing – original draft:** Nashid Islam, A. K. M. Motiur Rahman Bhuiyan, Afroja Alam, Mostofa Kamal Chowdhury, Jheelam Biswas, Md. Maruf Ahmed Molla, Mostofa Monwar Kowshik, Mridul Sarker, Nezamuddin Ahmed.

**Writing – review & editing:** Nashid Islam, A. K. M. Motiur Rahman Bhuiyan, Afroja Alam, Mostofa Kamal Chowdhury, Jheelam Biswas, Palash Chandra Banik, Md. Maruf Ahmed Molla, Mostofa Monwar Kowshik, Mridul Sarker, Nezamuddin Ahmed.

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
