## [Decision Letter · Decision Letter 0]

15 Sep 2022

PONE-D-22-12704Coping Strategy among the Women with Metastatic Breast Cancer Attending a Palliative Care Unit of a Tertiary Care Hospital of BangladeshPLOS ONE

Dear Dr. Biswas,

Thank you for submitting your manuscript to PLOS ONE. After careful consideration, we feel that it has merit but does not fully meet PLOS ONE’s publication criteria as it currently stands. Therefore, we invite you to submit a revised version of the manuscript that addresses the points raised during the review process.

We look forward to receiving your revised manuscript.

Kind regards,

Keisuke Suzuki, MD, PhD

Academic Editor

PLOS ONE

Journal Requirements:

Reviewers' comments:

Reviewer's Responses to Questions

**Comments to the Author**

1. Is the manuscript technically sound, and do the data support the conclusions?

Reviewer #1: No

Reviewer #2: Yes

2. Has the statistical analysis been performed appropriately and rigorously? 

Reviewer #1: No

Reviewer #2: Yes

3. Have the authors made all data underlying the findings in their manuscript fully available?

Reviewer #1: No

Reviewer #2: Yes

4. Is the manuscript presented in an intelligible fashion and written in standard English?

Reviewer #1: No

Reviewer #2: Yes

5. Review Comments to the Author

Reviewer #1: 1. Some description did not have reference to support, for example "In Bangladesh, late presentation of breast cancer is very common mostly because of social stigma and lack of awareness."

2. The significance of this study and the rational of this study need to be strengthen and clarified. Why you focus on metastatic breast cancer? why you recruited patients from palliative care unit? And how and why palliative care can impact on the coping of the patients? And what's the hypothesis in this study?

3. In the sample criteria, what's you mean "the indoor and outdoor facilities of the Department of Palliative Medicine"? And what's the metastatic breast cancer your mean??

4. The rational of calculating sample size is not clear, why the authors used changed acceptance to estimate the sample size, it is not examined in this study?

5. In the results about the characteristics of the samples, it is not clear "family history of malignancy"? you mean have other cancer or breast cancer? you mean stage 4 cancer??

6. The main purpose of this study is not clear, if the authors would like to know the better coping to improve depression or anxiety, the analysis needs to be control some confounding factors. However, form the current analysis, it can not answer this question.

Reviewer #2: This is a very interesting topic which analyse adaptive behavior of women confronted with a metastatic breast cancer. This analyse could allow to help some patients to accept their disease, better apprehend and improve their quality of life during this step of their life. The methodology is clear and well thought out.

6. PLOS authors have the option to publish the peer review history of their article (what does this mean?). If published, this will include your full peer review and any attached files.

Reviewer #1: No

Reviewer #2: No

---

## [Author Response · Author response to Decision Letter 0]

28 Oct 2022

Manuscript title: Coping Strategy among the Women with Metastatic Breast Cancer Attending a Palliative Care Unit of a Tertiary Care Hospital of Bangladesh 

Manuscript ID: PONE-D-22-12704

Reviewer#1’s comments to the authors-

Comment 1: 1. Some description did not have reference to support, for example "In Bangladesh, late presentation of breast cancer is very common mostly because of social stigma and lack of awareness."

Reply: Thank you for your comment. We have added the required reference, Reference no 3. 

Comment 2: The significance of this study and the rational of this study need to be strengthen and clarified. Why you focus on metastatic breast cancer? why you recruited patients from palliative care unit? And how and why palliative care can impact on the coping of the patients? And what's the hypothesis in this study?

Reply: Thank you for your comment. We have elaborated and clarified the significance of the study in the background section. As it is an observational study, there is no definite hypothesis. 

Changes in the text: 

Line 92-116: Coping strategies adopted by cancer patients are not fixed in the stone. It changes with the trajectory of the disease. As the disease progresses, many of these patients lean on acceptance and problem-focused coping, but some of them adopt negative coping like denial. Such coping strategies are associated with multiple outcomes, including impact on quality of life, depression and anxiety symptoms, understanding illness and prognosis, and care at the end of life [13]. 

Palliative care is an approach that improves the quality of life of patients and their families facing the problems associated with life-threatening illness. Studies demonstrate that palliative care integrated with oncology care, not only improves outcomes but also enhances coping in patients with advanced cancer. Since coping strategies are modifiable, palliative care aims to help the patients with advanced cancer adapting more positive coping strategies and improving their symptom burden, mental health, treatment outcome and quality of life [13]. Early consultation with palliative care also helps in lowering anxiety and depression among these patients [14]

Since the initiation of palliative care in Bangladesh, Department of Palliative Medicine of Bangabandhu Sheikh Mujib Medical University has been providing this care to the cancer patients and their families. According to their database almost 28% of these patients are suffering from metastatic (stage IV) breast malignancy (manually calculated). The goal of providing palliative care to these patients is not only to treat their physical symptoms, but also offering psychological, social and spiritual support. But we know a little regarding the adoption of coping strategies among these patients, which hinders us from offering them necessary support to cope with the disease and decrease their sufferings. So, this study aims to explore the different coping strategies adopted by the women with metastatic (stage IV) breast cancer attending our department and their relationship with the common mental health issues like anxiety and depression and to our best of knowledge, it is the first study to do so. 

Comment 3: In the sample criteria, what's you mean "the indoor and outdoor facilities of the Department of Palliative Medicine"? And what's the metastatic breast cancer your mean??

Reply: Thank you. Indoor facilities means where the patients were admitted in the palliative care ward of our department. Outdoor facility consists of outdoor consultation, lymphedema care etc, and the patients received their treatment from there in OPD basis. We took patients from both areas. We have also clarified in the methods section. 

Changes in the text: 

Line 123- 127 : Patients having stage IV breast cancer above 18 years of age, either admitted in the palliative care ward or attended the outdoor facility of the Department of Palliative Medicine, BSMMU, were included in our study. Those patients, who have any disorders like stroke, motor neuron disease etc that can clearly interfere with cognition, were excluded. 

Comment 4: The rational of calculating sample size is not clear, why the authors used changed acceptance to estimate the sample size, it is not examined in this study?

Reply Thank you for your comment. There is a slight mistake in this part of the methods section. We have actually included all the stage IV breast cancer patients who attended our department during the study period meeting our sample criteria in this study. The sampling technique has been clarified in the methods section. 

Changes in the text: 

Line 128-135: We used census method to determine the sample size of the study. All the stage IV breast cancer patients either admitted in the palliative care ward or attended the outdoor facility of the Department of Palliative Medicine, BSMMU during the study period were listed from the hospital register. According to the hospital register, total 103 stage IV breast cancer patients visited our department (manually calculated) during the study period. Among them, 27 patients were admitted in the palliative care ward, and 76 received treatment form the outdoor. Six of them had pre-existing cognitive impairment, and 2 of them refused to give consent. So our final sample size was 95. 

Comment 5: In the results about the characteristics of the samples, it is not clear "family history of malignancy"? you mean have other cancer or breast cancer? you mean stage 4 cancer??

Reply: Thank you for your comment. It is actually the family history of breast cancer of any stage. We have corrected the information in Table 1. 

Comment 6: The main purpose of this study is not clear, if the authors would like to know the better coping to improve depression or anxiety, the analysis needs to be control some confounding factors. However, form the current analysis, it can not answer this question.

Reply: Thank you. This study is an observational study. The main purpose of the study is to explore the coping strategies adopted by the patients at a single point of time, as well as presence of common mental health problems like anxiety and depression among them. However, we didn’t take history of any pre-existing mental illness of the patients which is one of the confounding factors and also a limitation of this study. We have clarified this limitation at the end of the discussion section. 

Changes in the text: 

Line 334- 342: However, our study has several limitations. One limitation is that, this study only focuses on the coping strategies adopted at a single point of time, so it doesn’t reflect the changes in the coping strategies along with the disease progression. In addition to that, some confounding factors had not been controlled. For instance, past psychiatric history or other psychological issues, which might precipitate anxiety and depression among these patients, were not explored. Another limitation is that, this study is conducted in a single center using a small, nonrandom purposive sampling technique, so the results cannot be generalized. For a better understanding of coping strategies adopted by advanced cancer patients, multi-center studies with greater sample size are required

Reply to reviewer#2’s comments to the authors:

Comment 1: What do the average of the different domains (acceptance, religion, emotional and instrumental support) correspond to? It’s not specify in the summary’s methodology. 

Reply: Thank you for your comment. We have re-analyzed the data, and presented the domains in median and interquartile range. Higher value median indicates the more commonly adopted coping strategy. It is mentioned below the table 3. 

Changes in the text:

Line 48-50: Most commonly used coping strategies by patients were: acceptance (median 10; IQR 10), religion (median 9; IQR 8-10) and instrumental support (median 9; IQR 6-10).

Line 216-222: Among the different pattern of coping strategies adopted by the patients the most commonly used strategies were- acceptance (median 10; IQR 10), religion (median 9; IQR 8-10), and instrumental support (median 9; IQR 6-10) closely followed by emotional support (median 8; IQR 7-10), active coping (median 8; IQR 7-10), planning (median 8, IQR 8-10) and venting (median 8; IQR 6-10). The lowest used coping strategies were humor (median 2; IQR 2-3) and substance use (median 2; IQR 2) followed by denial (median 3; IQR 2-7) and behavioral disengagement (median 3; IQR 2-5) (Table 3). 

Comment 2: Can you write clearly what domain are correlated with each other? Are all these domains (emotional support, active coping, planning, acceptance, behavioral disengagement, venting) correlated with religion?

Reply: Thank you. We did a correlation matrix to see the internal correlation among the domains. That is added in the results section of both abstract and result as well as in table 4. 

Changes in the text: 

Line 50-54 and Line 225-232: Significantly strong positive correlations were found between emotional and instrumental support (R=0.7; p= 0.01), planning and active coping (R=0.7; p=0.01), acceptance and active coping (R=0.5; p=0.01), denial and self distraction (R=0.6; p=0.01), behavioral disengagement and denial (R=0.5; p=0.01), venting, emotional and instrumental support (R=0.7; p=0.01). Significantly strong negative correlations were observed between active coping, planning and denial (R=-0.6; p=0.01), behavioral disengagement, acceptance and planning (R=-0.5; p=0.01) (Table 4). 

Comment 3:In the last line of summary “and” is repeated 2 times: “and and management…”

Reply: Thank you. The mistake has been corrected

Changes in the text: 

Line 58-60: All women suffering from breast cancer should be routinely screened and assessed for phychological distress and ensure early intervention and management to promote a better quality of life.

Comment 5:In line 64: Can you correct “Diagnosis of metastatic breast cancer it is a great shock”. The correct phrase would be “Diagnosis of metastatic breast cancer is a great shock”. -

Reply: Thank you. We have corrected the phrase. 

Changes in the text: 

Line 72: Diagnosis of metastatic breast cancer is a great shock for the patient.

Comment 6:In line 64 I would write the sentence like that: “Diagnosis of metastatic breast cancer is a great shock for the patient. Its treatment and side effects have tremendous social and psychological impacts on her.

Reply: Thank you for your query. We have corrected the mistake.

Changes in the text: 

Line 72-73: Diagnosis of metastatic breast cancer is a great shock for the patient. Its treatment and side effects have tremendous social and psychological impacts on her [4].

Comment 7: Line 68: You say: “In Bangladesh, 68 late presentation of breast cancer is very common mostly because of social stigma and lack 69 of awareness”. So, are you sure of the percentage of metastatic first diagnosis (6%)? It seems low.

Reply: Thank you for your query. Around the world 6% patients with breast cancer patients present with metastasis at the first diagnosis. It is the global data. However, in Bangladesh the exact number of breast cancer patients present with metastasis at the first diagnosis is unknown as there is no national cancer registry. But most of the clinicians experience the situation. We have rephrased the lines with the appropriate reference. 

Changes in the text: 

Line 67-71: Breast cancer has been reported to be the highest prevalent (about 19.3 per 100000 women) malignancy among Bangladeshi women between 15 to 44 years of age. Many of these patients present at the late stage mostly because of social stigmata and lack of awareness. Unfortunately, there is no national cancer registry in Bangladesh so the exact number of these patients remains unknown [3]. 

Comment 8: Globally the methodology is well explained. Line 133: a point is missing after “questionnaire” and “it” just after should be deleted. 

Reply: Thank you. We have corrected the line.

Changes in the text: 

Line 153-155: The third part contained the validated Bangla version of the ‘Hospital Depression and Anxiety Scale (HADS)’ questionnaire [16]. The part contained seven items that assess anxiety and seven items that assess depression. 

Comment 9: Line 172: “More than half of them (68.4%) had no family history of malignancy”: I would add “history of malignancy known”.

Reply: Thank you. We have corrected the line.

Changes in the text: 

Line 194-195: More than half of them (68.4%) had no known family history of breast malignancy. 

Comment 10: Table 1: Mean of age is already notify in the text. It’s not necessary to put it in the table. I would write there also “Family history of malignancy”. 

Reply: Thank you. We have corrected the errors.

Comment 11: Line 182: “patients” is repeated 2 times.

Reply: Thank you. We have corrected the line.

Changes in the text: 

Line 208: Nearly half of the patients (47.4%) were found to have no anxiety according to..

Comment 12: Line 224: Can you correct this phrase. It not grammatically correctly written. I would write “In our study women, suffering from metastatic breast cancer, most frequently used coping strategies like acceptance, religion, emotional support, instrumental support, and planning. But…”

Reply: Thank you. We have made some changes in this section, so the line has been completely deleted in the revised manuscript.

Comment 13: Line 226: Do you want to say “behavioral disengagement”? –

Reply: Thank u. We have corrected the spelling. 

Comment 14: Line 268: Can you reformulate this phrase because it is not correct. Can specify seem to have what?

Reply: Thank you. We have corrected the line

Change in the text: 

Line 305-306: Strategies like substance use, behavioral disengagement, and self-blame seem to have significantly less impact among our participants. 

Comment 15: Line 280 to 282: Can you separate in two sentences? 

Reply: Thank you. There are some corrections in the discussion section.

Change in the text: 

Line 320-322: It is evident from our study that the women who adopt active coping suffer less from depression. On the other hand, there is no correlation found between coping strategy and anxiety.

Comment 16: Line 284: Paragraph from “One interesting finding…” You can’t say patients with this ECOG develop or adopt such and such coping strategy, because you don’t know what is the order of the mechanism. Namely, is the coping strategy a consequence of ECOG or is ECOG a consequence of coping strategy? You can only say that Patients with such and such ECOG have such and such coping strategy. The analysis only shows the correlation between both but don’t specify what comes first.

Reply: Thank you. We have corrected the lines.

Changes in the text: 

Line 327-332: Positively focused coping strategies such as instrumental and emotional support, venting and positive reframing are more commonly adopted by the patients with better performance on the ECOG performance scale, while patients with poorer performance status seemed to lean on to the negative coping like behavioral disengagement, self-distraction and denial. Performance status is an essential prognostic factor for the survival of patients with breast cancer.

Comment 17: Line 296: Idem than paragraph line 284.

Reply: Thank you. We have corrected the lines.

Changes in the text: 

Line 334-336: However, our study has several limitations. One limitation is that, this study only focuses on the coping strategies adopted at a single point of time, so it doesn’t reflect the changes in the coping strategies along with the disease progression. In addition to that, some

Comment 18: Line 299: and is repeated two times.

Reply: Thank you. We have corrected the line

Changes in the text:

Line 344-349: In this study we have found that, the women who had positive coping strategies suffer less from mental health problems. These strategies help people to adapt with their disease over time. Women with breast cancer should be encouraged to use positive coping strategies to ensure better adherence to treatment and also discourage negative strategies like denial, behavioral disengagement or self distraction which can delay their physical and psychological management.

Dear Reviewers, we are grateful for your kind time and substantial review; we believe now the manuscript is more improved, which will satisfy you.

---

## [Editor Report · Decision Letter 1]

21 Nov 2022

Coping Strategy among the Women with Metastatic Breast Cancer Attending a Palliative Care Unit of a Tertiary Care Hospital of Bangladesh

PONE-D-22-12704R1

Dear Dr. Biswas,

We’re pleased to inform you that your manuscript has been judged scientifically suitable for publication and will be formally accepted for publication once it meets all outstanding technical requirements.

Kind regards,

Keisuke Suzuki, MD, PhD

Academic Editor

PLOS ONE
---

## [Editor Report · Acceptance letter]

3 Jan 2023

PONE-D-22-12704R1 

Coping Strategy among the Women with Metastatic Breast Cancer Attending a Palliative Care Unit of a Tertiary Care Hospital of Bangladesh 

Dear Dr. Biswas:

I'm pleased to inform you that your manuscript has been deemed suitable for publication in PLOS ONE. Congratulations! Your manuscript is now with our production department. 

Kind regards, 

on behalf of

Dr. Keisuke Suzuki 

Academic Editor

PLOS ONE